# Phenolic and Anthocyanin Compounds and Antioxidant Activity of Tamarillo (*Solanum betaceum* Cav.)

**DOI:** 10.3390/antiox9020169

**Published:** 2020-02-18

**Authors:** Tung Diep, Chris Pook, Michelle Yoo

**Affiliations:** 1School of Science, Faculty of Health and Environment Sciences, Auckland University of Technology, Private Bag 92006, Auckland 1142, New Zealand; tung.diep@aut.ac.nz; 2Riddet Institute, Centre of Research Excellence, Private Bag 11 222, Palmerston North 4442, New Zealand; 3The Liggins Institute, The University of Auckland, Private Bag 92019, Auckland 1142, New Zealand; chris.pook@auckland.ac.nz

**Keywords:** Tamarillo, phenolics, anthocyanins, antioxidant, chlorogenic acid, delphinidin-3-rutinoside

## Abstract

This study examined phenolics and anthocyanins present in Amber, Laird’s Large and Mulligan cultivars of tamarillo that were cultivated in Whangarei, Northland of New Zealand. Samples were further separated by their tissue types, peel and pulp. Using LC-MS/MS, twelve polyphenols were quantified and six (ellagic acid, rutin, catechin, epicatechin, kaempferol-3-rutinoside and isorhamnetin-3-rutinoside) were detected for the first time in tamarillo. Mulligan cultivar showed the highest amounts of phenolic and anthocyanin compounds and the highest antioxidant activity. Phenolic compounds were mostly synthesized from shikimic acid route, and chlorogenic acid dominated the profile regardless of cultivar and tissue types. Anthocyanin profile was dominated by delphinidin-3-rutinoside in pulp. Higher amounts of anthocyanins were detected in this study, which may be explained by favourable growth conditions (high light intensity and low temperature) for anthocyanin biosynthesis in New Zealand. Higher antioxidant activity and total phenolic content in peels than in pulps were found when assessed by Cupric Ion-Reducing Antioxidant Capacity (CUPRAC), Ferric Reducing Ability of Plasma (FRAP) and Folin–Ciocalteu assays, and a positive correlation (*r* > 0.9, *p* ≤ 0.01) between the three assays was observed. Current findings endorse that tamarillo has a great bioactive potential to be developed further as a functional ingredient with considerable levels of antioxidant compounds and antioxidant activity.

## 1. Introduction

Tamarillo (*Solanum betaceum* Cav.) is a fruit species of family *Solanaceae* genus *Solanum*, which is also known as tree tomato as its flesh closely resembles to that of the tomato [1]. As a subtropical fruit, tamarillo is mainly grown in warmer and sheltered areas of the North Island in New Zealand (Auckland and Hawkes Bay) [2]. New Zealand is one of the leading producers of tamarillo [3], with the main export markets including America, Australia, Hong Kong, Singapore and Japan. The ripe fruit turns to various colours (yellow, orange, red or purple) depending on the cultivars and exhibits a slightly bitter, sour and astringent taste with a unique aroma [4]. In New Zealand, tamarillo are available in yellow, red and purple-red cultivars, with the red being more popular and more common than the others. Tamarillo is considered as underutilized fruit due to its texture, strong flavour and some unidentified properties.

Phenolics (or polyphenols) and anthocyanins are secondary plant metabolites which are also known as antioxidants. Health-promoting effects arising from prolonged high intake of polyphenols have been reported extensively in the literature, where reduction in risks of certain cancers, cardiovascular diseases and diabetes and reduction in lipid oxidation and protective effects against free radicals have been observed. Phenolics and anthocyanins have been shown to possess significant antioxidant activity. For phenolic compounds, hydroxycinnamic acids (caffeic, ferulic, *p*-coumaric and rosmarinic acids); hydroxybenzoic acids (gallic and vanillic acids); flavonol (kaempferol) and flavanone (naringin) had been found in tamarillo from Ecuador, New Zealand [4] and Malaysia [5]. Anthocyanins are water-soluble pigments present in plants. They are commonly used as colourants in the food industry. Antioxidant activity of anthocyanins is able to protect other food ingredients against oxidation and help to maintain the nutritional value of foods. Presence of anthocyanins, such as cyanidin, delphinidin and pelargonidin rutinosides in tamarillos from Brazil [6], Colombia [7], Ecuador [4,8] and New Zealand [4], has been reported. The antioxidants are believed to partially contribute to health-promoting effects of tamarillo, including antioxidation and antioxidative stress [9], anti-obesity [10], anticancer [5,11] and anti-microbial activity [12], as well as protection against lipid oxidation [13]. Variations arising from environmental factors (temperature, light intensity and soil conditions) and post-harvest storage conditions contribute to variability in the presence of antioxidants and their activities in tamarillos. Often, these are dependent on where they are produced, even for the same cultivar. Scarce information on anthocyanin and phenolic compositions of tamarillo, especially for those from New Zealand, are currently available, or data are several decades old [1,14]. To our best knowledge, anthocyanin and phenolic profiles of tamarillo separated by tissue types, into peel and pulp, have never been systematically investigated. The pulp of the tamarillo is mostly consumed fresh, and the peel (skin) is discarded as waste.

In the present study, cultivars of Amber, Laird’s Large and Mulligan tamarillos grown in New Zealand were explored for their phenolic and anthocyanin compositions, as well as antioxidant activity, for the first time. Results from the current study will contribute to valorisation of tamarillo as a good source of antioxidants and to propose potential benefits of the by-product (peels).

## 2. Materials and Methods 

### 2.1. Materials

Three cultivars of ripe tamarillo fruit (21–24 weeks) were studied, including Amber (yellow), Laird’s Large (red) and Mulligan (rich purple-red) from Northland, New Zealand. For each cultivar, 15 fruits were randomly selected to obtain representative samples, washed with running water and then dried. Pulp and peel were manually separated, snap-frozen in liquid nitrogen, lyophilised and ground to powder. Samples were packed and stored at −20 °C until analysis.

### 2.2. Chemicals, Reagents and Standards

All chemicals and reagents used were AnalaR grade or better. Acetone, acetonitrile, acetic acid, methanol, isopropanol (IPA), formic acid, ethyl acetate, toluene and hydrochloric acid were obtained from Thermo Fisher (Auckland, New Zealand). Folin-Ciocalteu reagent; neocuporine; copper (II) chloride; 2,4,6-Tris(2-pyridyl)-s-triazine (TPTZ) and Trolox (6-Hydroxy-2,5,7,8-tetramethylchromane-2-carboxylic acid) were obtained from Sigma-Aldrich (Auckland, New Zealand). 

Analytical grade standards of polyphenols, including caffeic acid, (±) catechin, (−) epicatechin and *p*-coumaric acid, were also purchased from Sigma-Aldrich (Auckland, New Zealand). Analytical standards of other phenolics (gallic acid, ellagic acid, ferulic acid, chlorogenic acid, rutin, kaempferol, kaempferol-3-rutinoside and isorhamnetin-3-rutinoside) and four anthocyanins (cyanidin-3-glucoside, cyanidin-3-rutinoside, delphinidin-3-rutinoside and pelargonidin-3-rutinoside) were obtained from Extrasynthese (Genay Cedex, France). Milli-Q water was produced by Purite Fusion Milli-Q water purifying machine (Purite Limited, Thame, Oxon, UK).

### 2.3. Extraction and Quantification of Phenolics

A 50 ± 0.5 mg mass of dried, powdered sample was placed in an amber 1.8 mL glass vial, and a 1000 μL volume of 50% methanol was added. Samples were vortexed for 30 s and incubated in the dark for 30 min at 4 °C with vortexing every 5 min. The samples were then centrifuged at 10,000 RCF for 10 min at 4 °C to drive phase-transition. The supernatant was transferred to 1.5 mL polypropylene Eppendorf tube and centrifuged again at 10,000 RCF for 10 min at 4 °C. The extract was separated and transferred to a low-volume vial insert inside an amber 1.8 mL glass vial, then stored at −20 °C until analysis.

The LC-MS/MS consisting of an Agilent 1260 Infinity Quaternary LC System (Santa Clara, CA, USA) and an Agilent 6420 triple quadrupole mass spectrometer with multimode ionisation source (model G1978B) was used to identify phenolic compounds in pulp and peel of tamarillo cultivars. The phenolic compounds were separated along a Cortecs C18 column (2.1 × 100 mm, 2.7 μm; Waters, Tauton, Ireland). For mobile phase, 0.1% formic acid in Milli-Q for A and 0.1% acetic acid in acetonitrile for B were used. LC gradient was set at: 0–12 min, 97% of A; 12–13.5 min, 75% of A; 13.5–15.5 min, 10% of A and 15.5–23 min, 97% of A. Autosampler temperature was set at 4 °C. Injection volume and flow rate were 3.0 μL and 0.25 mL/min, respectively. Mass spectrometer (MS) was run in the negative mode with the total time of 23.0 min. The multimode (MMI) source operating at electrospray ionization (ESI) parameters were gas temperature of 325 °C, gas flow of 6 L/min. Capillary voltage and nebulizer were set to 2.0 kV and at 60 psi, respectively. Vaporizer temperature was set at 200 °C, and the column temperature was 25 °C. 

Preliminary results showed that the retention characteristics of the peaks suggested the presence of several phenolic compounds. Multiple reaction monitoring (MRM) was applied to detect and quantify these analytes. Optimal MRM transitions of target compounds, including retention time, precursor ion (*m*/*z*), product ion (*m*/*z*) and collision energy (V), are shown in Appendix A. Chromatograms of standard injections are presented in Figure 1. Calibration curves of analyte standards were constructed, and the linearity of these curves with at least six appropriate concentrations was used for method validation. A summary of the standard curves for each polyphenol with their associated regression equations, linear fit correlation coefficient (*R*^2^) and calibration range are presented in Appendix A. The sensitivity of the method was assessed by limit of detection (LOD) and limit of quantitation (LOQ). These two parameters were calculated based on the standard deviation of the response and the slope from the standard curve [15], and these are also presented in Appendix A. 

### 2.4. Extraction and Quantification of Anthocyanins 

Anthocyanins in the frozen, powdered tamarillo samples (50 ± 0.5 mg) were extracted with a mixture of 50 μL of formic acid, 200 μL of isopropanol (IPA) and 400 μL of Milli-Q. The sample was vortexed, followed by sonication in a water bath at 50 °C for 20 min. After addition of toluene (1.0 mL) into the solution to separate two liquid phases, the mixture was vortex-mixed again and centrifuged at 10,000 RCF for 10 min. The lower, aqueous phase (60–80 μL) was placed in a clean Eppendorf tube and centrifuged again at 10,000 RCF for 5 min. Then, 50 μL of the extract diluted with 50 μL of Milli-Q was placed into low-volume glass insert in a 1.8 mL brown glass autosampler vial. The sample was kept at −20 °C until analysis.

LC-MS/MS coupled with fluorescence detection (FLD) and XSelect C18 column (2.1 × 100 mm, 3.5 µm; Waters, Tauton, Ireland) were used to detect anthocyanin compounds. Flow rate, injection volume and total run time were 0.4 mL/min, 1.0 μL and 9.0 min, respectively. The column temperature was set at 25°C. The mobile phase A was 0.1% formic acid in acetonitrile and B was 0.6% formic acid in Milli-Q. The elution gradient was set at 0–4 min, 18% of A; 4–9 min, 6% of A. Double detection was implemented in a diode-array detector (DAD) at 520 nm as the preferred wavelength, and the MS was run in the negative ion mode. The optimised MS program conditions operating at ESI were gas temperature of 300 °C, gas flow of 6 L/min, vaporizer temperature of 300 °C, nebulizer gas at 60 psi and capillary voltage of 2.2–2.5 kV. 

Preliminary results showed that some of the peaks were coherent with cyanidin-3-glucoside, cyanidin-3-rutinoside, delphinidin-3-rutinoside and pelargonidin-3-rutinoside based on their retention characteristics by UV-spectrometry. Multiple reaction monitoring (MRM) transitions of four analyte standards were optimised by Agilent MassHunter Optimizer software (Santa Clara, CA, USA), and data were collected by Agilent MassHunter Acquisition software. Chromatograms of standard injections and optimal MRM parameters of analytes are shown in Figure 1 and Appendix A, respectively. Quantification of anthocyanins in tamarillo extracts were implemented using standard calibration curves fitted with at least six suitable concentrations (Appendix A). Method validation was carried out by assessing the linear fit correlation coefficient (*R*^2^) of the linearity of standard curve. The LOD and LOQ were also calculated based on the standard deviation of the response and the slope from the standard curve [15]. A summary of the method validation for all of the identified anthocyanins is presented in Appendix A. 

### 2.5. Total Phenolic Content (TPC)

Total phenolic content (TPC) of tamarillo was identified by the Folin-Ciocalteu method reported by Dorman, Koşar, Kahlos, Holm and Hiltunen [16], with some modifications. Firstly, 100 mg of lyophilised sample was placed in a 15 mL centrifuge tube, and then 4 mL of 50% methanol was added. The mixture was homogenised and left to rest for 60 min. The sample then was centrifuged at 1500 RCF for 15 min, and the extract from the top layer of the liquid was transferred into a 10 mL volumetric flask. Then, the residue was extracted for a second time with 4 mL of 70% acetone, and the steps described above, including homogenisation, resting and centrifugation, were followed. The extract was transferred to volumetric flask containing the first extract, and then, Milli-Q was added to the volumetric flask containing methanol and acetone extracts up to the 10 mL mark. From this, 1 mL from the volumetric flask was transferred to another volumetric flask and diluted to the 10 mL mark using Milli-Q. This was used as the extracted sample solution.

The extracted sample solution (1 mL) was placed into a 10 mL glass vial, and then, 500 μL of Folin-Ciocalteu reagent was added into the initial mixture and kept at room temperature for 5 min. 1.5 mL of 20% sodium carbonate (Na_2_CO_3_) solution was added to the vial, and the mixture was incubated at room temperature for 120 min in the dark. The extracted sample solution was transferred to a cuvette, and absorbance was measured at 765 nm against the blank using a UV-spectrophotometer (Ultrospec 7000, Cambridge, England). The results were achieved by interpolating the absorbances on a standard curve obtained with gallic acid (2.5–200 mg/L) under the same conditions. Results were expressed as mg of gallic acid equivalent (GAE) per 100 g of dry weight (DW).

### 2.6. Antioxidant Activity by Ferric Reducing Ability of Plasma (FRAP) and Cupric ion-Reducing Antioxidant Capacity (CUPRAC) Assays

FRAP assay was conducted according to Benzie and Strain [17], with modifications. Preparation of the sample extract solution was the same as described in the TPC protocol. FRAP reagent (1:1:10, *v*/*v*/*v*) was prepared by mixing FeCl_3_·6H_2_O (20 mM) with TPTZ (10 mM) in 40 mM HCl and acetate buffer (300 mM, pH 3.6). The sample extract solution (100 μL) was mixed with Milli-Q (900 μL) and the FRAP reagent (2 mL). Blank was prepared the same way, where the sample extract was replaced with Milli-Q. Absorbance was measured at 593 nm on a UV-spectrophotometer (Ultrospec 7000) after incubation at room temperature for 4 min. Quantification was carried out based on a Trolox standard curve, generated from 5 to 160 mg/L. The antioxidant capacity of the extracts was expressed as μmol Trolox equivalent antioxidant capacity per gram of dry weight (μmol TEAC/g DW).

The CUPRAC method published by Özyürek, Güçlü, Tütem, Başkan, Erçağ, Celik, Baki, Yıldız, Karaman and Apak [18] was used, with some modifications. Similar sample preparation procedure described in the TPC protocol was used. The sample extract solution (1 mL) was mixed with 1 mL of CuCl_2_ (0.01 M); 1 mL of ammonium acetate (1.0 M, pH = 7.0); 1 mL of Neocuporine (0.0075 M) in 96% ethanol and 0.1 mL of Milli-Q to achieve the total solution volume of 4.1 mL. The reaction was left for 5 min, and the absorbance was then measured at 450 nm on a UV-spectrophotometer (Ultrospec 7000) against MilliQ. A series of Trolox standard solutions in the range of 2.5–160 mg/L was used to produce a standard curve, and the CUPRAC values were reported in μmol TEAC/g DW. 

### 2.7. Statistical Analysis

Mean and standard deviation were calculated based on at least three independent measurements (*n* ≥ 3) for each experiment. Two-way analysis of variance (ANOVA) and Fisher’s (LSD) multiple comparison tests were applied to identify whether significant differences exist among different cultivars (Amber, Laird’s Large and Mulligan) and tissues (peel and pulp) of tamarillo, together with the interaction between these parameters. Pearson’s correlation coefficient was used to determine correlation among total phenolic content and the two other antioxidant assays. Data analysis was carried out using SPSS 25.0 (IBM Corp., Armonk, NY, USA), and the statistical significance level was set at *p* < 0.05. 

## 3. Results

### 3.1. Phenolic Compound Profiles

The LC-MS and the subsequent fragmentation of the predominant ion in MS-MS were used to identify phenolic compounds from the aqueous methanol extracts of tamarillo. As shown in Figure 1, twelve mixed phenolic standards were successfully separated in the negative ion mode, and further quantification of each identified polyphenol was carried out using a linear standard curve within a serial concentration range. The first peak was not ideal, but it did not influence accuracy and precision of the other compounds and the method. Good correlations of all the analysed phenolics were achieved with *R*^2^ of the linearity >0.99 (Appendix A). Additionally, the high sensitivity of the chromatography system and the method was confirmed through very low LOD and LOQ, which varied among different compounds. Rutin showed the lowest LOD and LOQ of 0.0092 and 0.0279 μg/L, respectively. Ellagic acid showed the highest LOD and LOQ of 1.7155 and 5.1984 μg/L, respectively (Appendix A). 

A total of 12 polyphenols were found and quantified in peel and pulp of three New Zealand grown tamarillo cultivars. Among these phenolics, six compounds had been previously reported in tamarillo [4,5,14]. Six other compounds were determined in tamarillo for the first time, and these include ellagic acid, rutin, catechin, epicatechin, kaempferol-3-rutinoside and isorhamnetin-3-rutinoside. Significantly (*p* < 0.05) different concentrations of phenolics were found between different cultivars and tissues, as shown in Table 1. In the current study, chlorogenic acid (3-caffeoylquinic acid) was the most abundant phenolic compound regardless of the cultivars and tissues. It ranged from 54.67 to 278.03 mg/100 g DW, with higher amounts present in Mulligan and lower amounts in Amber, as a general trend. Peels had more than three times of the chlorogenic acid concentration compared to the pulps. The presence of chlorogenic acid in tamarillo has been reported by Wrolstad and Heatherbell [14] and then later by Espin et al. [4] and Loizzo, Lucci, Núñez, Tundis, Balzano, Frega, Conte, Moret, Filatova and Moyano [19]. Espin et al. [4] also reported chlorogenic acid as the major phenolic compound in yellow and purple tamarillos from Ecuador and New Zealand, which agrees with the findings of the current study for Amber and Mulligan. Previously reported concentrations of chlorogenic acid in yellow and purple tamarillos from Ecuador was 25.04–42.73 and 50.33 mg/100 g DW, respectively, and in New Zealand purple cultivar, it was 163.62 mg/100 g DW [4]. This phenolic compound was also dominant in tamarillo from Colombia, with 25.38 mg/100 g DW in peel and 16.32 mg/100 g DW in pulp with seed [19]. These values were much lower than the current findings from New Zealand tamarillo. Another study reported that, in Ecuadorian tamarillo, the concentrations of caffeoylquinic acid and dicaffeoylquinic acid in the red type were 54.8 and 21.0 and, in yellow type, these were 32.8 and 17.1 mg chlorogenic acid equivalents per 100 g DW, respectively [8]. 

Another three hydroxycinnamic acids, caffeic acid, ferulic acid and *p*-coumaric acid were found in New Zealand tamarillos. Concentration of caffeic acid in peels was higher (approximately double) than that in pulps for all of the three cultivars examined. By contrast, *p*-coumaric acid showed higher concentration in pulps than in peels. The concentration of caffeic acid varied from 1.01 to 3.56 mg/100 DW, being the highest in Laird’s Large peel and the lowest in Amber pulp. In contrast, the Amber pulp had the highest concentration of *p*-coumaric acid (0.12 mg/100 g DW), while the Laird’s Large peel showed the lowest content of *p*-coumaric acid (0.02 mg/100 g DW). Caffeic and *p*-coumaric acids were quantified in pulp of Malaysian tamarillo, with concentration of 0.165 and 0.041 mg/100 g DW, respectively [5]. Loizzo et al. [19] quantified the concentration of *p*-coumaric acid in both peel and pulp of Colombian tamarillo at 0.02 mg/100 g DW. These values from both studies were lower than most of the current findings. Both Amber pulp and Laird’s Large pulp contained ferulic acid in extremely low concentrations (<0.005 mg/100 g DW), whereas all other samples had trace amounts ranging from 0.01 to 0.04 mg/100 g DW. Low concentrations (0.005 mg/100 g DW) of ferulic acid were also observed by Mutalib et al. [5]. A higher concentration of this acid has been observed in peel and pulp of tamarillo from Colombia with 0.87 and 0.76 mg/100 g DW, respectively [19]. Espin et al. [4] reported the presence of ferulic acid dehydrodimers in two yellow cultivars (0.12 and 0.06 mg/100 g DW) and the purple type tamarillo (3.27 mg/100 g DW) from Ecuador, as well as the purple-type tamarillo from New Zealand (21.17 mg/100 g DW), in their study. The higher contents of hydroxycinnamic acids in purple and red New Zealand tamarillos compared with the yellow one were in agreement with the finding observed for red and yellow tamarillos from Ecuador [8]. It was evident from our findings that hydroxycinnamic acids were one of the major polyphenolic groups in tamarillo and accounted for more than 85% and 55% of the polyphenols in peel and pulp of the three tamarillo cultivars, respectively (Figure 2). 

Two hydroxybenzoic acids, gallic acid and ellagic acid were found in New Zealand-grown tamarillo, which ranged from 0.9 to 1.09 mg/100 g DW (0.3–0.9%) (Table 1 and Figure 2). Both hydroxybenzoic acids showed low concentrations in all analysed samples, though significant differences between cultivars were observed (Table 1). The content of gallic acid was similar in both peel and pulp of Amber cultivar (approximately 0.8 mg/100 g DW), while the pulp of Laird’s Large and Mulligan had slightly higher gallic acid content than their peels (0.93 and 1.0 mg/100 g DW compared with 0.79 and 0.8 mg/100 g DW, respectively) (Table 1). Gallic acid was firstly quantified (0.302 mg/100 g DW) in Malaysian tamarillo by Mutalib et al. [5]. Despite being present in trace amounts (about 0.1 mg/100 g DW), the presence of ellagic acid was detected in tamarillo for the first time by the current study. All three cultivars in both peel and pulp showed relatively similar concentrations of ellagic acid, 0.09 to 0.12 mg/100 g DW (Table 1). There was no significant difference (*p* > 0.05) in ellagic acid content among three cultivars, and no significant interaction between cultivars and tissues was found. 

Apart from the two phenolic acid groups, one flavonol (kaempferol) was found in New Zealand-grown tamarillo at a very low concentration, from 0.43 to 0.5 mg/100 g DW. All the peels had similar concentrations of kaempferol, whereas the Amber pulp showed a slightly higher amount of this phenolic than the Laird’s Large and Mulligan pulps, with significant differences observed (Table 1). The percentage of this flavonol in the total phenolic concentration was 0.1–0.2% in peel and 0.4–0.5% in pulp (Figure 2). The concentration of this flavonol in Malaysian tamarillo has been reported as 0.05 mg/100 g DW [5], which was lower than the value obtained in the current study. 

Two flavanols, catechin and epicatechin, were reported for the first time in the current study. These accounted for 0.7–1.3% and 1.7–5.4% in peels and pulps of all phenolics, respectively (Figure 2). Amber showed significantly higher concentration of catechin in both peel (2.13 mg/100 g DW) and pulp (3.91 mg/100 g DW) than the two other cultivars. A low amount of catechin was detected in Laird’s Large and Mulligan cultivars (0.28–0.33 mg/100 g DW), as well. Significant differences (*p* < 0.05) were observed in catechin content among cultivars and tissues, with significant interactions observed between cultivars and tissues (Table 1). By contrast, the Mulligan had the highest content of epicatechin in both peel (2.6 mg/100 g DW) and pulp (2.31 mg/100 g DW) (Table 1). Additionally, peels and pulps of Amber and Mulligan cultivars showed relatively similar concentration of this flavanol, whereas the Laird’s Large peel had slightly lower amounts of epicatechin than the pulp (Table 1). There were no significant differences (*p* > 0.05) in epicatechin content between pulps and peels, and no significant interaction between cultivars and tissues was found. 

Presence of flavonol glycosides has never been reported in tamarillo before. Flavonol glycosides comprised 8.3–12.5% of all phenolics in peels and 35.3–41.8% of all phenolics in pulps (Figure 2). Rutin, known as quercetin-3-rutinoside, dominated the flavonol glycoside profile for peels of Amber and Mulligan cultivars by being the second-most abundant polyphenol (Table 1). The concentrations of rutin were significantly different among all tamarillo cultivars and tissues (*p* < 0.05), with higher concentrations observed in peels than in pulps (approximately 9.6, 7.5 and 3.8 times more for Mulligan, Amber and Laird’s Large cultivars, respectively). Among different tissues, Amber showed the highest content of rutin, with 24.33 and 3.23 mg/100 g DW in peel and pulp, respectively (Table 1). The current study reports kaempferol-3-rutinoside as the second-most abundant phenolic in pulps of all cultivars, as well as in Laird’s Large peel. Significant differences in the content of kaempferol-3-rutinoside were observed among six samples (*p* < 0.05) (Table 1). Laird’s Large pulp and Amber peel showed the highest and the lowest concentrations of kaempferol-3-rutinoside, with 50.04 and 8.32 mg/100 g DW, respectively. For pulp, it was approximately 2.6–3.6 times higher than that in the peel across all the cultivars (Table 1). Trace amount of isorhamnetin-3-rutinoside was found in pulp of all cultivars (≤0.16 mg/100 g DW). Among all peel samples, Laird’s Large had the highest concentration (0.96 mg/100 g DW), followed by Mulligan (0.9 mg/100 g DW) and then Amber (0.59 mg/100 g DW) (Table 1). 

The total concentration of phenolic compounds detected in the peels was slightly more than a double of the pulps for all cultivars, as shown in Table 1. For both peel and pulp, Mulligan showed the highest total phenolics (TPs). The TPs in Laird’s Large pulp were slightly higher than the Amber pulp, whereas the Amber peel and Laird’s Large peel had relatively similar TPs (Table 1). Previous findings in Colombian tamarillo showed that TPs in peel and pulp were 28.41 and 20.72 mg/100 g DW [19], which were lower than our findings. 

### 3.2. Anthocyanin Compound Profiles

Four anthocyanins were captured and quantified in peels and pulps of New Zealand-grown tamarillos using LC-MS/MS. All data acquired for method validation are shown in Appendix A and Appendix A with *R*^2^ > 0.999, as well as low LOD and LQD values. A representative MRM transitions chromatogram of the determined anthocyanins in tamarillo extracts and their anthocyanin profiles are illustrated in Figure 1 and Table 1, respectively. To our knowledge, this was the first attempt to quantify anthocyanins in two different tissues of tamarillo. Cyanidin-3-glucoside was quantified for the first time in Laird’s Large and Mulligan cultivars, although these were only detected in the peels, at 0.33 and 1.97 mg/100 g DW, respectively. Pulps showed higher total anthocyanin concentration than the peels in all cultivars. Additionally, Laird’s Large and Mulligan cultivars had higher concentrations of each anthocyanin than Amber for both peel and pulp tissues (Table 1). The proportion of different anthocyanins was relatively similar in Laird’s Large pulp and Mulligan pulp.

There was a huge variation in the total anthocyanin content of peels. For Mulligan, Laird’s Large and Amber, 259.18, 155.82 and 1.24 mg/100 g DW were found, respectively. Compared to the peels of other fruits with similar colour (red), the concentration of total anthocyanins was evidently higher in tamarillo. For example, the peels of red grape, red plum and red apple had the total anthocyanin concentrations of 27, 20 and 12 mg/100 g FW, respectively (approximately equal to 138.75, 156.62 and 83.1 mg/100 g DW, respectively) [20]. Total anthocyanin content in pulp of Mulligan, Laird’s Large and Amber was much higher than its counterpart (peel), with 486.84, 481.37, and 29.70 mg/100 g DW, respectively. The total anthocyanins in the pulp of tamarillo were higher compared to the pulp of strawberry, grape and cranberry, with 41.09, 28.09 and 19.30 mg/100 g FW, respectively (approximately equal to 454.03, 144.35 and 152.21 mg/100 g DW, respectively) [21]. 

Significant differences in individual and total anthocyanin concentrations were found among different cultivars and tissues. The interaction between cultivars and tissues were significant (*p* < 0.05) for anthocyanins (Table 1). Presence of cyanidin 3-rutinoside, delphinidin-3-rutinoside and pelargonidin 3-rutinoside have been previously reported in tamarillo [4]. Delphinidin-3-rutinoside was the most dominant anthocyanin in Laird’s Large and Mulligan pulps (254.76 and 273.36 mg/100 g DW, respectively), followed by pelargonidin 3-rutinoside and cyanidin 3-rutinoside. For peels of Laird’s Large and Mulligan, cyanidin-3-rutinoside was the most abundant anthocyanin; 68.72 and 114.47 mg/100 g DW, respectively (Table 1). Previously, cyanidin 3-rutinoside has been reported as the main pigment in peel of New Zealand purple variety [14], and delphinidin-3-rutinoside being the major anthocyanin in the pulp of the purple variety [4,14]. For Laird’s Large and Mulligan, delphinidin-3-rutinoside and pelargonidin-3-rutinoside showed significantly higher concentrations in pulps than in peels by 5.6–7.8 and 2.6–3.7 times more, respectively. By contrast, peels had higher concentrations of cyanidin-3-rutinoside than pulps by 2.0 and 3.7 times for Laird’s Large and Mulligan, respectively (Table 1).

Compared to Mulligan and Laird’s Large, which carry purple and red colours, Amber cultivar, carrying yellow colour, possessed the least amounts of anthocyanins. To date, the presence of anthocyanins has never been reported in New Zealand Amber cultivar or yellow variety from other sources. The major anthocyanin in Amber peel and pulp was pelargonidin-3-rutinoside (0.52 mg/100 g DW) and delphinidin-3-rutinoside (29.17 mg/100 g DW), respectively (Table 1). Concentrations of cyanidin-3-rutinoside, delphinidin-3-rutinoside and pelargonidin-3-rutinoside in Laird’s Large and Mulligan tamarillo in this study were significantly higher than that reported in the red variety from Ecuador [8] and in the purple cultivars from Ecuador and New Zealand [4]. Additionally, the New Zealand Laird’s Large and Mulligan tamarillos had greater total anthocyanin content (481.37 and 486.84 mg/100 g DW, respectively) than Brazilian tamarillo, with 8.5 mg/100 g FW (approximately equal to 70.83 mg/100 g DW, with estimated moisture of 88%) [6]; Ecuadorian tree tomato, with 165.1 mg/100 g DW [8] or 102.35 mg/100 g DW [4] and New Zealand purple cultivar from the previous study (168.88 mg/100 g DW) [4]. 

### 3.3. Total Phenolic Content (TPC)

As shown on Table 2, peels had significantly higher (*p* < 0.05) TPC and antioxidant capacity than pulps. From the three cultivars studied, Mulligan was the most bioactive (*p* < 0.05), with relatively high TPC (2225.06 and 874.09 mg GAE/100 g DW), as well as strong antioxidant activities, as determined by CUPRAC assay (265.29 and 71.57 μmol TEAC/g DW) and FRAP assay (161.74 and 72.14 μmol TEAC/g DW) for peel and pulp, respectively (Table 2). 

The TPC of peels and pulps varied among tamarillo cultivars, where the values were almost a double in peels compared to that of the pulps. The phenolic content was the highest in Mulligan peel (2225.06 mg GAE/100 g DW) and the lowest in Amber pulp (678.98 mg GAE/100 g DW). The purple and red cultivars have been reported to have greater values of phenolic contents than the yellow ones [1,8,22]. The TPC of red and gold tamarillos from New Zealand were 190.8 and 116.6 mg GAE/100 g FW (1564 and 1060 mg/100 g DW), respectively [1]. These were higher than the values in the current study. Vasco et al. [22] reported that the TPC of golden-yellow and purple-red tamaillo cultivars from Ecuador were 78 and 113 mg GAE/100 g FW (557 and 1413 mg/100 g DW), respectively. 

### 3.4. Antioxidant Activity

There are several assays to assess antioxidant activity in vitro, where CUPRAC and FRAP methods are the most widely used. Significant differences (*p* < 0.05) in the antioxidant activity were found among different cultivars and tissues, individually. A significant interaction between the cultivars and tissues was also observed. Antioxidant capacity, presented by the CUPRAC value, was significantly higher in Mulligan than in Laird’s Large and Amber cultivars for peels (265.29 compared with 136.68 and 117.59 μmol TEAC/g DW, respectively) and pulp (71.57 compared to 52.42 and 42.92 μmol TEAC/g DW, respectively) (Table 2). The CUPRAC values for peels were approximately 3.7, 2.8 and 2.6 times higher than pulps for Mulligan, Amber and Laird’s Large cultivars, respectively. 

Similar phenomena were observed when assessed with the FRAP assay. Mulligan showed the strongest anti-radical efficacy in both tissues, and peel owned higher antioxidant ability than pulp for all of the cultivars. The FRAP values varied from 52.23 to 161.74 μmol TEAC/g DW, with the highest value found in Mulligan peels and the lowest in Amber pulp. The FRAP values for pulp were 1.6–2.2 times less than that for the peels. Espin et al. [4] have reported the FRAP value for pulp of purple tamarillo from New Zealand as 50 μmol TEAC/g DW [4], which was lower than the current study. They have also reported the FRAP values for the pulp of yellow and purple tamarillos from Ecuador as 10–17 and 15 μmol TEAC /g DW, respectively [4]. 

### 3.5. Correlation between TPC, CUPRAC and FRAP

Results from Pearson’s correlation coefficient revealed a strong correlation between TPC and antioxidant activity. The correlation between TPC–CUPRAC and TPC–FRAP were 0.941 (*p* < 0.01) and 0.906 (*p* < 0.01), respectively (Table 3). The high correlation between TPC and antioxidant activity (determined by DPPH method) of tamarillo had been previously reported by Acosta-Quezada, Raigon, Riofrio-Cuenca, Garcia-Martinez, Plazas, Burneo, Figueroa, Vilanova and Prohens [23] for *r* = 0.8607 and by Mutalib et al. [11] for *r* = 0.998. Vasco, Ruales and Kamal-Eldin [24] reported the correlation between the TPC and FRAP value of 17 Ecuadorian fruits, including purple-red and golden-yellow tamarillos, as 0.62. Similarly, the results of the two antioxidant assays (CUPRAC and FRAP) were highly correlated (*r* = 0.959, *p* < 0.01) (Table 3). For 17 Ecuadorian fruits, including tamarillo, a correlation of between DPPH and FRAP assays was 0.908 [24]. 

## 4. Discussion

Three tamarillo cultivars from New Zealand were analysed for phenolic and anthocyanin components and assessed for antioxidant capacity. In the literature, the characteristics of Amber and Mulligan cultivars, which are also known as yellow and purple varieties, respectively, are rarely found. Most of the reported studies in tamarillo have focused on Laird’s Large cultivar, which is also known as red variety, as it is the most commonly grown and consumed tamarillo cultivar.

Variations in the profiles of phenolic and anthocyanin compounds in tamarillo between the current study and the previously reported values in the literature may be partly explained by climate and environmental factors, including stronger UV light, dry soils and lack of rain in Northland of New Zealand where the tamarillos for the current study were sourced from. Clean and green environment of New Zealand and their breeding and pest management techniques may have provided ideal conditions for tamarillo cultivation, compared to other countries. Variations may also have risen from the cultivars of fruit, post-harvest handling, storage period, ripeness of the fruit and extraction and analytical methods for quantification. For example, Mertz et al. [8] used aqueous acetone containing 2% formic acid as extraction solvent for anthocyanin quantification, and Espin et al. [4] used a combination of methanol and 0.1% formic acid for extraction with 0.1% trifluoracetic acid and acetonitrile as mobile phases. In this study, a mixture of formic acid, Milli-Q, IPA and toluene was used as extraction solvent, with a mobile phase of 0.1% formic acid in acetonitrile and 0.6% formic acid in Milli-Q. A toxicity problem can be caused by using organic solvents (methanol or ethanol) for anthocyanin extraction [25]; therefore, water-based extracts was used. Additionally, we used pure formic acid to lower the pH of the solution at which these anthocyanins are more stable. The formation of flavylium cation at acidic condition is known to significantly enhance the solubility of anthocyanins in water [25]. IPA is a good nontoxic alternative to preserve biological specimens; in this case, anthocyanins. Toluene was used to separate different layers to enhance the purity of the extracts. The use of LC-MS/MS for quantification of these compounds may have helped in identifying those at lower concentrations. 

As summarised in Figure 3, most of the phenolic compounds found in tamarillo might be synthesised from the shikimic acid route. In general, phenolic compounds in plants are synthesised from this route [26], where carbohydrate precursors are converted to phenylalanine and then to *trans*-cinnamic acid, *p*-coumaric acid and dihydroflavonols. With the addition of hydroxyl groups and sugars and methylation, these compounds are further derived to form various secondary phenolic compounds, which were observed in tamarillo from the current study. For example, addition of sugar molecules to the flavonol skeletons produce kaempferol-3-rutinoside, rutin and isorhamnetin-3-rutinoside (Figure 3), and these were detected in this study (Table 1). 

Chlorogenic acid, which is an ester formed between caffeic acid and quinic acid (Figure 3), functions as an intermediate in lignin biosynthesis. In *Solanaceous* species which include eggplant, potato and tomato, chlorogenic acid has been reported as the main soluble phenolic compound [27], which was also observed as the dominating phenolic compound in the current study. The concentrations of chlorogenic acid in New Zealand tamarillos was significantly higher than that in tomato pulp, which ranged from 24.58 to 53.10 mg/100 g DW [28]. Variations in genetic traits and physiological characteristics of plants (e.g., height of the tree and its respective absorption of sunlight) may explain the differences in chlorogenic acid between these two species. According to Clé, Hill, Niggeweg, Martin, Guisez, Prinsen and Jansen [29], significantly higher levels of soluble phenolics are found in plants when grown under natural sunlight. With the strong UV light of New Zealand, this may have helped in producing higher amounts of phenolic compounds to those sourced from other countries. Ripeness of the fruit is also known to influence the production of chlorogenic acid [30]. According to Onakpoya, Spencer, Thompson and Heneghan [31], intakes of foods containing high levels of chlorogenic acid can result in significant reductions in systolic and diastolic blood pressures. Health benefits of chlorogenic acid include anti-diabetic, anti-carcinogenic, anti-inflammatory, anti-obesity and antioxidant properties [32], which would further increase the bioactive potential for tamarillo as a functional ingredient.

Rutin, a natural flavone derivative, was the second-most abundant polyphenol identified in New Zealand-grown tamarillos. Pure rutin carries yellow or yellow-green colour, which may explain its higher presence in Amber cultivars than in two other varieties, regardless of the tissue type. Rutin is a powerful antioxidant, known for its capacity to strengthen the blood cell walls [33], with antitumor, antibacterial and antiviral [34], antioxidant and anticarcinogenic properties [35]. Moreover, rutin has been reported to help in collagen production and to enhance utilisation of vitamin C [36], which is present in significant amounts in tamarillo (25 mg/100 g) [37]. Jang, Wang, Lee, Lee and Lim [38] reported that kaempferol-3-rutinoside, known as a powerful α-glucosidase inhibitor, owned anti-adipogenic potential, which may act as an anti-obesity agent. These authors indicated that kaempferol-3-rutinoside showed inhibitory effects on adipogenesis to 48.2% without cytotoxicity. 

Caffeic and gallic acids possess biological functions associated with the modulation of carcinogenesis. Gallic acid has been reported as a protector against oxidative damage and chemo-preventive agents, resulting in the death of several tumour cell lines [5]. Foods derived from flavonols—in this case, kaempferol—exhibit multiple health-promoting effects, such as inhibiting the formation of cancer cells. Flavanols such as catechin and epicatechin, which were determined in tamarillos for the first time, could contribute to increasing the overall antioxidant power of tamarillo. Catechin is known to possess anticancer, anti-obesity, anti-inflammatory and neuroprotective properties. With the presence of these phenolics, bioactive values of tamarillo are compatible to other super fruits. 

On top of health benefits, anthocyanins have raised a growing interest in improving postharvest handling. With the presence of anthocyanins, lessening of overripening and longer shelf-life has been observed [39], and this may explain the long season of tamarillo. Anthocyanins enhance the antioxidant activity of fruits by suppressing reactive oxygen species (ROS), which will subsequently slow down the overripening process [40]. By controlling the ROS burst, anthocyanins improve fruit resistance to botrytis, minimising fungal growth [40]. Current findings support the previous literature that delphinidin-based anthocyanins are the abundant structure of *Solanaceous* species, including pepper, tomato, eggplant and potato [39]. Delphinidin-based anthocyanins have been reported as inhibiting thrombosis and reducing vascular inflammation [41]. Additionally, by preventing keratinocyte apoptosis, it is able to protect human skin against UV-B irradiance [39]. 

Cyanidin and its glycosides have been recognised as strong antioxidants. Their antioxidant activity is stronger than that of vitamin E, vitamin C and resveratrol [25]. To date, bioavailability of cyanidin-3-glucoside has been the most explored among these major anthocyanins [25]. Czank, Cassidy, Zhang, Morrison, Preston, Kroon, Botting and Kay [42] have observed that within 48 h of ingestion, the relative bioavailability of cyanidin-3-glucoside reached 12.38%, in which 5.37% was excreted in urine and 6.91% was lost through breathing. This anthocyanin is easily absorbed into the plasma [25]. Usually, cyanidin is concentrated in the skin of fruits [43], which was also observed in the current study (Table 1). All of the peel samples showed higher amounts of cyanidin-3-rutinoside than the pulps. Cyanidin-3-glucoside were only detected in Laird’s Large and Mulligan peels. Therefore, anthocyanins extracted from the peels of these tamarillos could be used as food additives. Similar applications have been made with grape skins to produce E163 [25]. The current findings also raise an interest in using anthocyanin as a natural colorant to replace synthetic food dye. 

In nature, cyanidin appears as a reddish-purple pigment, while delphinidin is seen as a purple pigment. This can explain why the Amber cultivar had the least anthocyanin, compared to Laird’s Large and Mulligan. Formation of flavylium cations are predominant in red pigments of anthocyanins, and these are stable at acidic conditions [25]. The pH of golden-yellow and purple-red tamarillo cultivars are 3.2–3.5 and 3.5–3.6, respectively [22], which are suitable for these anthocyanins to be stable. The richer concentrations of anthocyanins in pulps than in peels might be due to the presence of seeds, which have dark purple colours. Hurtado, Morales, González-Miret, Escudero-Gilete and Heredia [44] reported that delphinidin-3-rutinoside owned the greatest ability in capturing the 2,2′-azino-bis(3-ethylbenzothiazoline-6-sulphonic acid), known as ABTS radical, followed by cyanidin-3-rutinoside and then pelargonidin-3-rutinoside.

Light and temperature during the growth of tamarillos have been reported to influence the anthocyanin metabolism. Anthocyanin biosynthesis is known to be stimulated by high light intensity [45], and it is also affected by light quality in most plants. Guo, Han and Wang [46] reported that UV-A radiation enhanced anthocyanin contents in tomatoes, compared to white light. Anthocyanin accumulation in *Solanaceae* species is induced by low temperatures [47,48]. The strong UV light of New Zealand and average temperatures of 10–18 °C from April to October, when tamarillo are being grown, may explain the significantly higher amounts of anthocyanins in New Zealand tamarillos compared to other sources.

Peels exhibited higher total phenolic contents, as determined by the Folin–Ciocalteu method, and stronger antioxidant activity than the pulp of tamarillo. This seemed common in fruit, in general. Polyphenols found in peels have been reported to possess antifungal and antibacterial properties [12]. Extraction and purification of these compounds into a functional ingredient or additive may be applicable for the food and pharmaceutical industries and to help with utilisation of by-product (peels). High antioxidant activity of the peels is also related to higher colour stability [44], which is helped by the presence of phenolic acids, organic acids, flavones, flavanones, flavonols and flavanols [49,50]. 

The significant difference of antioxidant capacity observed across the cultivars and the tissues could be related to their phenolic compositions. From the current study, Mulligan (purple) cultivars showed the highest antioxidant capacity. Red cultivar showed higher antioxidant capacity than the gold cultivar from New Zealand, which is in agreement with Lister et al. [1] who used the ABTS assay. Greater antioxidant activity in purple-red tamarillos compared to the yellow species from New Zealand were similar to Ecuadorian tree tomatoes, which were assessed by the ORAC assay [8] and DPPH assay [24]. Many studies have demonstrated that tamarillo possesses higher antioxidant capacity than commonly consumed fruits, such as apples, oranges, red grapes [8], kiwi fruit, pineapple [1], tomatoes and cherry tomatoes [51], although the concentrations of phenolic compounds in tamarillo are similar or lower than these fruits. This may be due to the presence of certain phenolic compounds or other components which own stronger antioxidation properties; for example, anthocyanins, carotenoids and phenolic acids. As discussed, the presence of phenolics, anthocyanins, TPC and antioxidant activity of tamarillo suggest that this fruit, particularly the peels, has a great potential to be utilised as a functional ingredient. Antioxidant activity of tamarillo has also been evaluated using in vitro cancer cell lines. With 3-(4,5-dimethylthiazol-2-yl)-2,5-diphenyltetrazolium (MTT) assay, Ordóñez, Cardozo, Zampini and Isla [52] demonstrated that tamarillo reduced oxidative stress in HepG2 liver cancer cells and provoked apoptosis in a dose-dependent manner. Prevention of proliferation and viability of the HepG2 liver cancer cell line and MDA-MB 231 breast cancer cell line have also been observed through MTT assay by Mutalib et al. [11], where the most significant effect was seen with application of the crude ethanol extracts from tamarillo containing 2.53 mg GAE/g DW of TPC. From our study, TPC in the aqueous ethanol extracts of peels and pulps were much higher, as shown in Table 2 (6.79 to 22.3 mg GAE/g DW). Therefore, we assume that similar cytotoxic effects may be expected if the tamarillo extract was to be applied to HepG2 and MDA-MB 231 cell lines. Presence of higher TPC in peels compared to pulps may suggest a new use of the peels, which are often discarded as by-products. Mutalib et al. [11] further reported that the cytotoxic effects of tamarillo extracts against MDA-MB 231 and HepG2 cells were as good as that of doxorubicin, which is a commercial anticancer drug. The strong correlation observed between TPC and antioxidant activity (as identified by CUPRAC, FRAP and DPPH methods) indicated that polyphenols were the major antioxidants in tamarillo regardless of the cultivars and tissues.

Chlorogenic acid, caffeic acid, catechin, ellagic acid, rutin and kaempferol and its glucosides were among those with validated antioxidant activity demonstrated through in vitro and in vivo studies. Treatments with chlorogenic acid of up to 40μM (14.2 μg/mL) have shown a protective ability against *t*-BOOH-induced ROS generation in HepG2 cells (*p* < 0.05) [53] and H_2_O_2_-induced oxidative stress in human HaCaT cells (*p* < 0.05) [54] and DNA damage in MCF-7 and MAD-MB-231-cultured human breast cancer cells [55] and in human blood lymphocytes (*p* < 0.001) [56]. Through supplementation of chlorogenic acid (5 mg/kg body weight) in diabetic rat models, reduction of lipid hydroperoxide formation (*p* < 0.05) has also been reported [57]. From the current study, chlorogenic acid found in aqueous ethanol extracts of peels and pulps from New Zealand tamarillos ranged from 113.2 to 133.6 and 26.4 to 36.3 μg/mL, respectively. Treatment effects were seen at lower chlorogenic concentrations in the previous studies; hence, similar protective effects may be observed with these tamarillos.

Extensive studies related to bioactive polyphenolic compounds and their related functions in humans are found in the literature. These include inhibitory effects against oxidative stressors (*t*-BHP, H_2_O_2_ and FeSO_4_) in PC12 cells by ellagic acid [58]; protective effects against DNA damage in MCF-7 and MAD-MB-231-cultured human breast cancer cells by caffeic acid at 20 μM [55]; delay in lipid oxidation and reduction in the formation of malondialdehyde activity (MDA) of platelets in human plasma (*p* < 0.05) by catechin at 20 to 200 μg/mL [59]; reduction of ROS generation in H_2_O_2_-treated APPswe cells (*p* < 0.05) by rutin at 50 to 100 nM [60]; prevention of concanavalin A (ConA)-induced activation of T cell proliferation and nitric oxide and ROS formation in LPS-induced RAW 264.7 macrophage cells by kaempferol and its glycosides [61]. The data obtained from the current study show that the tamarillos possessed higher amounts of those aforementioned compounds and their effective concentrations. Therefore, similar effects may be observed if further in vitro and/or in vivo studies were conducted with these tamarillos. 

Similarly with the anthocyanins, high antioxidant capacity has also been demonstrated in human colon cancer (Caco-2), human hepatocarcinoma (HepG2), human endothelial (EA.hy926) and rat vascular smooth muscle (A7r5) cells with anthocyanins from bilberries and blueberries at low concentrations (EC_50_ < 1 μg/L, nM range) [62]. An excellent antioxidant capacity has been observed in the plasma of rats (*p* < 0.01) [63] and in the serum of humans (*p* < 0.05) [64] with increased intakes of anthocyanins. Furthermore, treatment with anthocyanins extracted from purple-tomato peel at 52.5 μg/mL showed reduction in ovarian cancer cell lines (*p* < 0.05) [65]. Provided that the total anthocyanins contents in the extracts of Laird’s Large peel and Mulligan peel were 87 and 144 μg/mL, respectively (Table 1; unit converted to μg/mL), tamarillo peels are likely to show similar extents of antioxidant activity. The respective ratio between these antioxidant compounds and anthocyanins may further enhance the antioxidant capacity of the fruit, compared to the application of a single compound to the cell lines. Further research of in vitro cell line-based assays and animal models should be implemented for a deeper understanding of antioxidant activity in tamarillo.

To our best knowledge, this was the first study to measure the TPs and TPC of three tamarillo cultivars from New Zealand separated into peel and pulp tissues. There was a slight difference between TPC identified by the Folin-Ciocalteu method and TPs calculated from all phenolics determined by LC-MS/MS. This discrepancy may be due to the Folin-Ciocalteu method measuring the TPC-containing free phenolic groups only while the LC-MS/MS detects phenolics containing glycosylated or ester-linked groups, such as rutin, kaempferol-3-rutinoside or isorhamnetin-3-rutinoside. Additionally, the total concentrations of polyphenols calculated from the LC-MS/MS results were only based on the identified compounds, while the TPCs were measured for the entire extracts.

## 5. Conclusions

Examination of antioxidant compositions and their capacities in three New Zealand tamarillo cultivars showed that tamarillo peels possessed higher amounts of phenolic compounds, total phenolic content and antioxidant activity than the pulps. Pulps had higher anthocyanins concentration than the peels. For cultivars, Mulligan showed the highest antioxidant activity and the highest amounts of phenolic and anthocyanin compounds, in general. Phenolic profile was dominated by chlorogenic acid regardless of cultivars and tissues. The anthocyanin profiles of pulp in all three cultivars were dominated by delphinidin-3-rutinoside. Antioxidant capacity of tamarillos exhibited relatively high values and were strongly correlated with high total phenolic content. Presence of these bioactive compounds highlight the potential of tamarillo for further utilisation in food and pharmaceutical industries.

Further research in tamarillo to explore the interaction of these phenolics and anthocyanins with other food components and how their antioxidant activity is influenced during food processing would be advantageous. Tamarillo remains underutilised despite its bioactive components. It has a relatively long fruit season, and they are relatively shelf-stable compared to other fruits (e.g., strawberries), which would be of great advantage. 

## Figures and Tables

**Figure 1 antioxidants-09-00169-f001:**
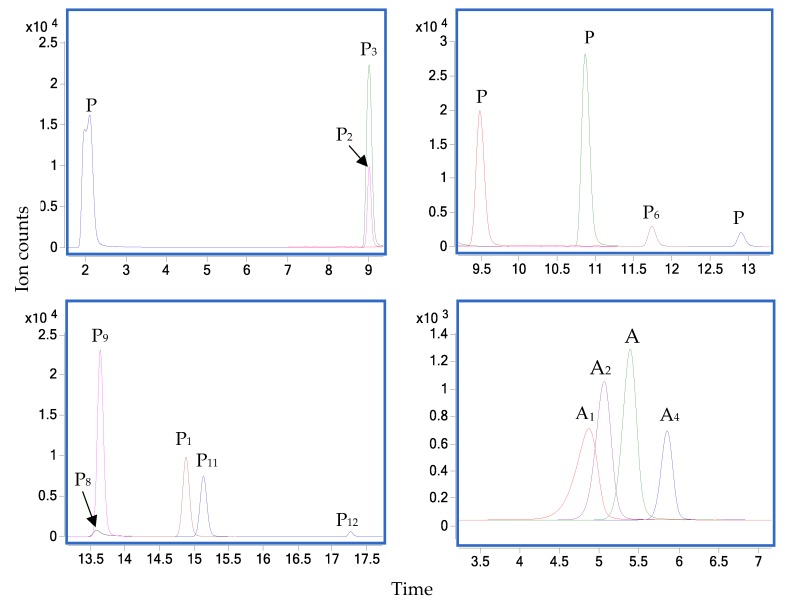
Multiple reaction monitoring (MRM) chromatograms of 12 phenolics (P) and 4 anthocyanins (A) standards. P_1_: gallic acid, P_2_: chlorogenic acid, P_3_: catechin, P_4_: caffeic acid, P_5_: epicatechin, P_6_: *p*-coumaric acid, P_7_: ferulic acid, P_8_: ellagic acid, P_9_: rutin, P_10_: kaempferol-3-rutinoside, P_11_: isorhamnetin-3-rutinoside, P_12_: kaempferol, A_1_: dephinidin-3-rutinoside, A_2_: cyanidin-3-glucoside, A_3_: cyanidin-3-rutinoside and A_4_: pelargonidin-3-rutinoside.

**Figure 2 antioxidants-09-00169-f002:**
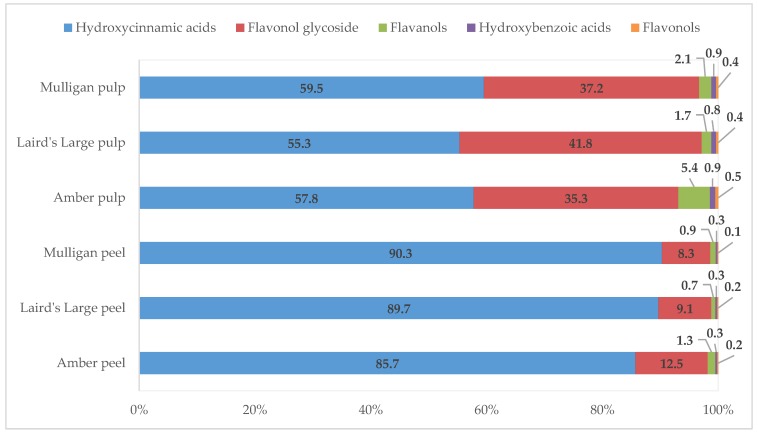
Proportion (%) of five dominating phenolic classes separated by tissue and cultivar types of tamarillo.

**Figure 3 antioxidants-09-00169-f003:**
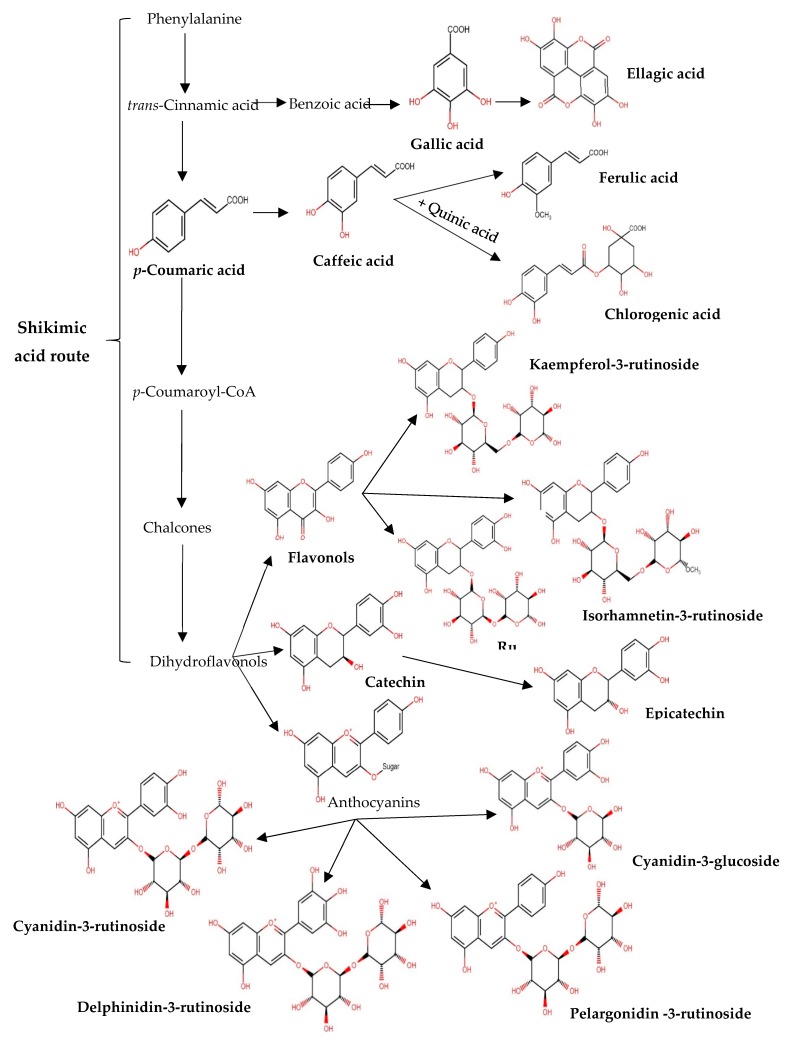
Outline of phenolics and anthocyanins biosynthesis from phenylalanine in tamarillos.

**Table 1 antioxidants-09-00169-t001:** Profiles of phenolics and anthocyanins (mg/100 g DW) in three tamarillo cultivars, separated by cultivars and tissues. Values are expressed as Mean ± SD (*n* = 4).

Bioactive Compounds	Peel			Pulp			*p*-Value *		
	Amber	Laird’s Large	Mulligan	Amber	Laird’s Large	Mulligan	C	T	C × T
Chlorogenic acid	225.85 ± 12.43^ax^	231.18 ± 9.76^bx^	278.03 ± 11.89^cx^	54.67 ± 3.81^ay^	66.35 ± 1.1^by^	73.95 ± 1.98^cy^	<0.05	<0.05	<0.05
Caffeic acid	2.61 ± 0.18^ax^	3.56 ± 0.52^bx^	3.52 ± 0.85^bx^	1.01 ± 0.03^ay^	1.2 ± 0.06^by^	1.32 ± 0.04^by^	<0.05	<0.05	0.0328
*p*-coumaric acid	0.05 ± 0.01^ax^	0.02 ± 0.01^bx^	0.03 ± 0.01^cx^	0.12 ± 0.01^ay^	0.07 ± 0.01^by^	0.08 ± 0.01^cy^	<0.05	<0.05	<0.05
Ferulic acid	0.03 ± 0.03^ax^	0.01 ± 0.01^bx^	0.04 ± 0.03^cx^	<0.005^ay^	<0.005^by^	0.02 ± 0.02^cy^	<0.05	<0.05	*0.1158*
Total hydroxycinnamic acids	228.53 ± 12.65^ax^	234.77 ± 10.3^bx^	281.63 ± 12.78^cx^	55.80 ± 3.86^ay^	67.62 ± 1.18^by^	75.37 ± 2.04^cy^	<0.05	<0.05	<0.05
Gallic acid	0.8 ± 0^ax^	0.79 ± 0^bx^	0.8 ± 0.01^bx^	0.79 ± 0^ay^	0.93 ± 0.06^by^	1 ± 0.2^by^	<0.05	<0.05	<0.05
Ellagic acid	0.12 ± 0.04^x^	0.11 ± 0.03^x^	0.1 ± 0.04^x^	0.11 ± 0.02^y^	0.09 ± 0.01^y^	0.09 ± 0.02^y^	*0.2304*	0.0483	*0.6023*
Total hydroxybenzoic acids	0.91 ± 0.05^ax^	0.9 ± 0.03^abx^	0.91 ± 0.05^bx^	0.90 ± 0.03^ay^	1.01 ± 0.07^aby^	1.09 ± 0.22^by^	0.0227	<0.05	0.0147
Kaempferol	0.43 ± 0.01^ax^	0.43 ± 0.01^bx^	0.43 ± 0.01^bx^	0.5 ± 0.04^ay^	0.45 ± 0.01^by^	0.45 ± 0.01^by^	<0.05	<0.05	<0.05
Total flavonols	0.43 ± 0.01^ax^	0.43 ± 0.01^bx^	0.43 ± 0.01^bx^	0.5 ± 0.04^ay^	0.45 ± 0.01^by^	0.45 ± 0.01^by^	<0.05	<0.05	<0.05
Catechin	2.13 ± 0.83^ax^	0.28 ± 0^bx^	0.33 ± 0.02^bx^	3.91 ± 0.68^ay^	0.3 ± 0^by^	0.32 ± 0^by^	<0.05	<0.05	<0.05
Epicatechin	1.36 ± 0.03^a^	1.49 ± 0.12^b^	2.6 ± 0.72^c^	1.34 ± 0^a^	1.73 ± 0.01^b^	2.31 ± 0.02^c^	<0.05	*0.763*	*0.0545*
Total flavanols	3.49 ± 0.86^ax^	1.78 ± 0.13^bx^	2.94 ± 0.74^cx^	5.25 ± 0.68^ay^	2.03 ± 0.01^by^	2.63 ± 0.02^cy^	<0.05	<0.05	<0.05
Rutin	24.33 ± 1.85^ax^	3.71 ± 0.45^bx^	12.68 ± 1.1^cx^	3.23 ± 0.08^ay^	0.97 ± 0.03^by^	1.32 ± 0.03^cy^	<0.05	<0.05	<0.05
Kaempferol-3-rutinoside	8.32 ± 0.57^ax^	19.22 ± 1.2^bx^	12.45 ± 1.09^cx^	30.72 ± 0.97^ay^	50.04 ± 1.12^by^	45.6 ± 1.41^cy^	<0.05	<0.05	<0.05
Isorhamnetin-3-rutinoside	0.59 ± 0.05^ax^	0.96 ± 0.05^bx^	0.9 ± 0.02^cx^	0.16 ± 0.01^ay^	0.13 ± 0^by^	0.14 ± 0^cy^	<0.05	<0.05	<0.05
Total flavonol glycosides	33.25 ± 2.47^ax^	23.89 ± 1.7^bx^	26.03 ± 2.21^bx^	34.11 ± 1.06^ay^	51.14 ± 1.14^by^	47.06 ± 1.44^by^	<0.05	<0.05	<0.05
Total phenolics	266.62 ± 16.04^ax^	261.77 ± 12.16^bx^	311.93 ± 15.80^cx^	96.56 ± 5.67^ay^	122.26 ± 2.42^by^	126.60 ± 3.74^cy^	<0.05	<0.05	<0.05
Delphinidin-3-rutinoside	0.43 ± 0.02^ax^	32.41 ± 2.98^bx^	49.11 ± 2.23^cx^	29.17 ± 2.47^ay^	254.76 ± 6.33^by^	273.36 ± 12.7^cy^	<0.05	<0.05	<0.05
Cyanidin-3-glucoside	n.d	0.33 ± 0.04^a^	1.97 ± 0.14^b^	n.d	n.d	n.d	<0.05	–	–
Cyanidin-3-rutinoside	0.29 ± 0^ax^	68.72 ± 4.33^bx^	114.47 ± 5.97^cx^	0.18 ± 0.02^ay^	25.94 ± 1.99^by^	30.67 ± 2.76^cy^	<0.05	<0.05	<0.05
Pelargonidin-3-rutinoside	0.52 ± 0.06^ax^	54.36 ± 3.24^bx^	93.63 ± 2.48^cx^	0.35 ± 0.03^ay^	200.66 ± 8.51^by^	182.81 ± 11.17^cy^	<0.05	<0.05	<0.05
Total anthocyanins	1.24 ± 0.08^ax^	155.82 ± 10.58^bx^	259.18 ± 10.81^cx^	29.70 ± 2.52^ay^	481.37 ± 16.83^by^	486.84 ± 26.63^cy^	<0.05	<0.05	<0.05

n.d: not detected. * Statistical significance for cultivar (C), tissue (T) and the interaction of both types (C **×** T). Means shown in ^a, b, c^ are significantly different at *p* < 0.05 between cultivars. Means shown in ^x, y^ are significantly different at *p* < 0.05 between tissues. SD values of less than 0.004 are presented as 0.

**Table 2 antioxidants-09-00169-t002:** Total phenolic content and antioxidant activity of tamarillo separated by cultivars and tissues. Values are expressed as Mean ± SD (*n* = 4).

Tissues	Cultivars	Total Phenolic Content (mg GAE/100 g DW)	CUPRAC Value (μmol TEAC/g DW)	FRAP Value (μmol TEAC/g DW)
Peel	Amber	1583.8 ± 40.09^ax^	117.59 ± 9.35^ax^	84.7 ± 11.13^ax^
	Laird’s Large	1673.28 ± 63.97^bx^	136.68 ± 6.72^bx^	102.55 ± 12.19^bx^
	Mulligan	2225.06 ± 50.87^cx^	265.29 ± 18.35^cx^	161.74 ± 14.53^cx^
Pulp	Amber	678.98± 19.09^ay^	42.92 ± 8.73^ay^	52.23 ± 6.7^ay^
	Laird’s Large	707.04 ± 30.65^by^	52.42 ± 8.39^by^	60.19 ± 5.21^by^
	Mulligan	874.9 ± 30.48^cy^	71.57 ± 7.81^cy^	72.14 ± 9.41^cy^
*p*-value *	C	<0.05	<0.05	<0.05
	T	<0.05	<0.05	<0.05
	C × T	<0.05	<0.05	<0.05

* Statistical significance for cultivar (C), tissue (T) and the interaction of both types (C × T). Means shown in ^a,b,c^ are significantly different at *p* < 0.05 between cultivars. Means shown in ^x,y^ are significantly different at *p* < 0.05 between tissues. FRAP: Ferric Reducing Ability of Plasma and CUPRAC: Cupric Ion-Reducing Antioxidant Capacity.

**Table 3 antioxidants-09-00169-t003:** Correlation between total phenolic content (TPC) and antioxidant activity (measured by CUPRAC and FRAP assays) of three tamarillo cultivars.

Content		TPC	CUPRAC	FRAP
TPC	Pearson’s Correlation	1	0.941 **	0.906 **
	Sig. (2-tailed)		0.000	0.000
	Number	24	24	24
CUPRAC	Pearson’s Correlation	0.941 **	1	0.959 **
	Sig. (2-tailed)	0.000		.000
	Number	24	24	24
FRAP	Pearson’s Correlation	0.906 **	0.959 **	1
	Sig. (2-tailed)	0.000	0.000	
	Number	24	24	24

** Correlation is significant at the 0.01 level (2-tailed).

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
