# Peer review of "Phenolic and Anthocyanin Compounds and Antioxidant Activity of Tamarillo (Solanum betaceum Cav.)"

_antioxidants, 2020, doi:10.3390/antiox9020169_

Round 1

Reviewer 1 Report

The paper was conducted rigorously from a phytochemical point of view. The results are consistent with the conclusion. 

The authors discussed about the antioxidant effects of Tamarillo.

This is also highlighted in the title. 

In my opinion, it would be better define these as antiradical effects. 

Additionally, a pharmacological evaluation, aimed to confirm the putative antioxidant role, would surely add value to the study. The major revision decision was substantiated by the lack of a specific antioxidant test in a selected biological model, namely in vitro cell or tissue paradigm.

Reviewer 2 Report

In the present study, phenolics and anthocyanins present in fruit pulp and peels of three tamarillo cultivars were evaluated.

Overall, the manuscript is very well prepared and addresses the topic in an excelent manner.

Correct the title: "Cav." should not be written in Italics.

Line 140: correct formatting of the text and rephrase to "were coherent".

Correct the heading to "Results and "Discussion". A soon as references are cited in the text then this section refers to discussion also.

Line 300: correct to "has never been reported".

Round 2

Reviewer 1 Report

The manuscript was improved after revision. 

Therefore it can be accepted.